# Entrepreneurial Intention of Chinese Students Studying at Universities in the Community of Madrid

**Susana Lin** [1,*]**, Carmen De-Pablos-Heredero** [2] **, José Luis Montes Botella** [3] **and Cristina Lin-Lian** [3]

1   Faculty of Legal and Social Sciences, Rey Juan Carlos University, Paseo de los Artilleros, s/n, 28032 Madrid, Spain
2   Department of Business Economics (Administration, Management and Organization), Applied Economics II and Fundamentals of Economic Analysis, Rey Juan Carlos University, Paseo de Los Artilleros s/n, 28032 Madrid, Spain; carmen.depablos@urjc.es
3   Department of Applied Economy I, Rey Juan Carlos University, Paseo de Los Artilleros s/n, 28032 Madrid, Spain; joseluis.montes@urjc.es (J.L.M.B.); cristina.lian@urjc.es (C.L.-L.)
*   Correspondence: s.lin.2016@alumnos.urjc.es

**Abstract:** Entrepreneurial intention is one of the most representative indicators of entrepreneurship action. The Chinese community has an increasing presence in the community of Madrid both in the educational field and in the business field. This paper studies the relationship between different socioeconomic and cultural variables and the entrepreneurial intention of Chinese students studying at universities in the community of Madrid. As a methodology, an analysis based on the application of structural equation modeling (SEM) has been chosen, since it is an exploratory analysis where this type of data has not been previously identified. The results show that subjective norms, the perception of control, and the motives of entrepreneurship have a positive and significant relationship with entrepreneurial intention. In contrast, the attitude toward entrepreneurship, gender, previous work experiences, and the existence of entrepreneurial parents do not have a significant relationship with entrepreneurial intention.

**Keywords:** entrepreneur; entrepreneurship; entrepreneurial intention; Chinese students; community of Madrid; entrepreneurship education

## 1. Introduction

Entrepreneurship is very important for a country's development and economic growth since it generates employment, innovation, and wealth [1]. For this reason, it is necessary to understand how the entrepreneur's mind works and the habits and reactions to what surrounds him or her, as well as study how to meet his or her needs to accelerate the creation of companies, job innovation, and economic and social wealth. Entrepreneurship is a complex activity since it depends on personal, sociodemographic, psychological, economic, political, and cultural factors. Studying all these factors together will help create an effective predictor of entrepreneurial intent [2].

Intention is the best indicator of future execution of any planned conduct, especially if the action is not standard, difficult to detect, or involves unforeseeable delays, namely in the intention to set up companies [3]. Analyzing samples composed of university students has also been highlighted, since they are considered potential entrepreneurs compared with other samples [2,4].

Spain presents high potential for business development, as shown in the Total early-stage Entrepreneurial Activity (TEA), which measures the companies created in the last three and a half years. Traditionally, it has moved between 5 and 6%, below the European average, with that in the year 2020 being 8.1% according to the data published by the Global Entrepreneurship Monitor (GEM) report [5]. The low rate of entrepreneurial activity in Spain can be explained by, among other factors, cultural aspects, since the Spanish

population tends to present a greater fear of failure compared with other countries due to its conservative culture [6]. In 2020, Spain ranked first in terms of the perception of fear of failure as the main obstacle to entrepreneurship, with a percentage of 64% compared with the 47% European average. In addition, the perception of opportunities in the market for the incorporation of new companies was 16.5% in 2020, which was a decrease compared with previous years, falling far below the European average (40.5%) and placing the country in the last position within the high-income countries. All of this is linked to the fact that the primary motivation of entrepreneurship is to earn a living because there is no work, increasing from 47% to 72% from 2019 to 2020, which are reasons that can explain the low TEA rate [5].

In the case of China, it presented a TEA of 8.7% in 2019, which represented a significant decrease compared with previous years and was caused by the pretensions of nationalist protectionism that emerged in 2019 and the beginning of COVID-19. Despite this decline, the rate of entrepreneurial activity in China is still above the European average. In addition, 79.3% of the Chinese population perceive entrepreneurship as an excellent professional option, compared with 53.7% in the case of the Spanish population. This could be due to, among other factors, the fact that most of the Chinese population—79.4% to be precise—perceives good opportunities in the market to start a business, which is an essential precedent for carrying out the entrepreneurial process, placing itself in one of the countries that has best valued this perception and with a rate well above the average of middle-income economies (51.6%). Regarding the fear of failure, 43.9% perceive it as a major obstacle to entrepreneurship, standing slightly below the average of middle-income economies (44.6%). Within the indicators of culture, 80.2% of the Chinese population perceives that equity in the living standards of society is desirable, placing it as one of the countries that has best valued this factor and well above the average of 67.4% [7].

As for the recent actions to strengthen entrepreneurship in Spain, we can highlight the "Spain Entrepreneurial Nation Strategy", presented in February 2021 by the President of the Government, and the investments and reforms to be supported with the funds from the Recovery, Transformation, and Resilience Mechanism of the European Union, which favor above all the promotion of the entrepreneurial ecosystem [8].

In the case of China, regarding the initiatives developed by the government for the promotion of entrepreneurship, a plan called "Period of the 14th Five-Year Plan (2021–2025)" is being carried out, which is intended to massively promote entrepreneurship and innovation, better implement the requirements of the new development concept, stimulate the vitality of the market to boost development, expand employment, and benefit the livelihoods of citizens. Through this, great support was offered to employment by administrations, authorities, and banking entities, especially university graduates, promoting new engines of growth and favoring the internal dynamics of economic development [9].

It should be noted that the Spanish entrepreneurial ecosystem (conditions of the environment for entrepreneurship in Spain) is within the top 20, above the European average due mainly to a progressive strengthening of government policies, education training, and financial support for entrepreneurship [5]. The communities where it is easier to start a business within Spain are Madrid, Catalonia, and Valencia due to the wide range of training programs and greater digitalization [10].

Thanks to the efforts made by both Spain and China for the internationalization of university education, a progressive increase in the number of Chinese students in Spanish universities has been achieved, reaching a maximum of 12,571 Chinese students in the academic year 2019–2020, and within these, 42.35% did so in universities in the community of Madrid.

Therefore, Madrid is the city with the highest concentration of Chinese students due to its high number of universities, many of which have made the largest number of agreements with China [11,12].

The reasons why Chinese students come to Spain to study are, among others, to learn or improve their level of Spanish, recommendations from friends, and to know the Spanish

culture. In addition, they think that a Spanish degree has additional prestige and will offer them more professional opportunities [13].

In this work, an analysis will be carried out on the different dimensions that affect the entrepreneurial intentions of Chinese students who are carrying out university studies in the community of Madrid through SEM analysis based on the data obtained in surveys carried out on a representative sample.

This paper aims to know the entrepreneurial intention of Chinese students present at universities in the community of Madrid through the study of different variables and their influence on entrepreneurial intention and the influence of entrepreneurship education on the development of entrepreneurship in universities, as well as an assessment of the different training activities for the development of the entrepreneurial spirit. The sociodemographic factors that favor the creation of entrepreneurial intention will also be studied, and an assessment of the different motives for entrepreneurship in the Chinese community will be executed.

China is characterized by an entrepreneurial culture, so this work aims to know what the variables that determine the entrepreneurial intention of the Chinese community and the motives for entrepreneurship are.

This paper will help to know the reason why the Chinese community has so much interest in establishing commercial relations with Spain. Recognizing the determinants of entrepreneurial intention will allow us to recognize students with the entrepreneurial potential to motivate and offer support in the entrepreneurship process. Knowing the most valued training activities to promote the entrepreneurial spirit will allow for designing a more effective syllabus for the development of entrepreneurial intention in universities. As for the study of the sociodemographic factors of the sample, it will allow us to recognize why the Chinese community has a greater intention of entrepreneurship than the Spanish profile.

After the introduction, in Section 2, the hypotheses will be raised. In Section 3, the methodology used to contrast the hypotheses formulated will be detailed. In Section 4, the results obtained in the paper will be described. In Section 5, the discussion and conclusions will be presented, encompassing in the latter the limitations and some of the possible future lines of research.

## 2. Literary Review

### 2.1. Entrepreneurial Intention

Intention plays a crucial role in human behavior [14]. Most behaviors that influence society, such as new business creation or health-related behaviors, are carried out through voluntary control [15]. There are empirical evidence that intention is the best predictor of everyone's behavior [16,17].

We can define entrepreneurial intention as a cognitive representation that encompasses the actions to be developed by the individual for the creation of new companies and greater value for existing companies [18].

Entrepreneurship ideas begin with inspiration, but intentions are what make such ideas manifest in the mind of the individual [19]. Therefore, individuals do not set up a company by reflex but do it intentionally, since it is a planned action from which we can distinguish a series of steps, events, or factors that lead to the creation of a company. This is the intention the first step of this process, and from this, the form and direction of the company to be constituted is determined [20,21].

We can differentiate three stages in the entrepreneurship process [22], as shown in Figure 1.

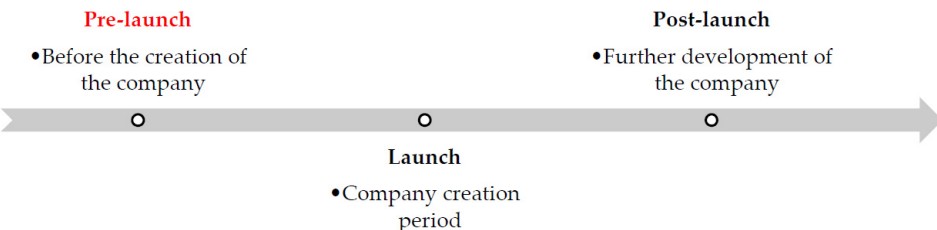

**Figure 1.** Stages of the entrepreneurship process. Own elaboration from Moriano (2005).

Entrepreneurial intention is in the pre-launch phase and is the most studied aspect of the process by various authors [23]. It has been proven that intention is the best indicator of future execution of any planned behavior, especially if the action is not common, difficult to detect, or involves unforeseeable delays, exactly as in the case of the intention to create companies, so behavioral intention models are suitable in this type of research [3].

Thus, entrepreneurial intention is considered a prerequisite to entrepreneurial behavior and the most important antecedent to acting in incorporating a new company [24,25]. Numerous studies of entrepreneurial intention have been conducted with university students because of the importance of university education in the creation of entrepreneurship [26], but this type of study has not previously been conducted with Chinese students from universities in Madrid, who have an increasing presence in both the labor and educational fields, which is the reason for our study.

It has been empirically demonstrated that entrepreneurial intention is determined by variables that can be grouped into personal factors, which distinguish entrepreneurs from other individuals, and environmental aspects, among which we can find the sociodemographic characteristics of individuals [25].

Numerous models try to study the aspects and stages of entrepreneurial intention, such as Ajzen's theory of planned behavior [17], a model used by numerous authors to study the entrepreneurial intention of individuals.

*2.2. Hypotheses*

2.2.1. Theory of Planned Behavior (TPB)

It starts from the idea that all behaviors arise after planning, through which the intention to carry out the behavior can be predicted [17].

The determinants of intention that then lead to behavior are the following [17].

Attitude. This is the favorable or unfavorable assessment of the person toward the behavior. These arise from the beliefs and opinions of the individual [17]. Several authors found that entrepreneurial intention is determined by the attitudes and beliefs of the individual toward entrepreneurship [27,28]. Hence, a positive belief in entrepreneurship greatly increases the entrepreneurial intention of the individual.

In contrast, a negative belief is a barrier to the generation of entrepreneurial intention, which then influences behavior [29]. The attitude can also be reflected in the attributes that the individual considers about the entrepreneurial action, such as the satisfaction that the individual perceives when performing a certain behavior, which would stimulate the entrepreneurial intention or, otherwise, the individual can interpret the venture as a higher monetary expense and greater personal dedication, which would decrease entrepreneurial intention [30,31].

In addition, individuals with more outstanding entrepreneurial attitudes perceive greater capacities to detect new opportunities in the market and are more willing to assume the risks derived from creating new companies [32]. They are also willing to devote more time and dedication to the entrepreneurial process [33]. Based on the previous arguments, we raise H1:

**Hypothesis 1 (H1):** *The attitude of the university student toward entrepreneurship is positively related to his or her entrepreneurial intention.*

Subjective norms. These are the degree by which the action fulfills the desires of people and are essential to the being who is carrying them out, such as family expectations and recommendations from friends [17]. Several studies have shown a positive relationship between the subjective norms that the individual perceives and his or her entrepreneurial intention [28,34]. This dimension is also studied in the entrepreneurial potential model. The subjective norm is considered, together with the attitudes of the individual, as the determinants of desirability. These have a direct and positive relationship with the entrepreneurial intention [35]. Regarding subjective norms, we will formulate the following hypothesis:

**Hypothesis 2 (H2):** *The subjective norms of the university student toward entrepreneurship are positively related to his or her entrepreneurial intention.*

Perception of behavior control. This is the degree of difficulty the person perceives to carry out the action, taking into account their abilities [17]. When the individual perceives that he or she has the necessary skills to create his or her own company and considers the process viable, that is when he or she will be encouraged and begin to carry it out [19,30]. The self-efficacy of the individual fundamentally determines this dimension to develop entrepreneurial action [36]. Self-efficacy is conditioned by the skills and abilities that the individual possesses toward a behavior, and these variables are especially relevant in educational entrepreneurship. Entrepreneurial universities are the place where students are provided with the skills and abilities necessary for the creation of new companies [37]. Previous studies have also shown that the perception of control is positively related to entrepreneurial intention [27,38].

Based on what has been described, we will formulate the following hypotheses:

**Hypothesis 3 (H3):** *The perception of control of the behavior of the university student toward entrepreneurship is positively related to his or her entrepreneurial intention.*

**Hypothesis 4 (H4):** *The abilities of the university student are positively related to the perception of control of the university student.*

### 2.2.2. Entrepreneurship Education

Education in entrepreneurship is key to the success of entrepreneurial activity, since entrepreneurs often have to face problems that need multidisciplinary training [39]. Several authors confirmed that entrepreneurship education significantly impacts entrepreneurship [38,40,41]. On the one hand, entrepreneurship education improves entrepreneurial self-efficacy [42,43] through experiences of mastery, vicarious experience, verbal persuasion, and emotional activation, increasing students' perceptions of viability [19,43]. On the other hand, the skills and attitudes toward entrepreneurship perceived by the individual are reinforced through entrepreneurship education. This is what is called entrepreneurship know-how, and various studies have shown the contribution of education to the development of entrepreneurial attitudes and, within these, especially inspiration [44]. Entrepreneurship education is also related to social norms, playing a fundamental role in the socialization of people during their studies in universities [35], thus increasing the desirability of business creation [45]. Numerous studies have empirically shown that entrepreneurship education positively impacts entrepreneurial intention [41,46].

Regarding entrepreneurship education, we will formulate the following hypothesis:

**Hypothesis 5 (H5):** *Entrepreneurship education is positively related to entrepreneurial intention in university students.*

### 2.2.3. Sociodemographic Factors

Gender

According to the GEM report, men have historically tended to present a greater entrepreneurial intention. However, this difference has been decreasing in recent years, presenting an evolution of constant growth from 2005, when the TEA was 7.2% and 4.2% in men and women, respectively, to 2020, with a TEA of 5.6% and 4.8%, respectively [5]. Along the same lines, we can find jobs in which it has been shown that men tend to have a greater entrepreneurial intention compared with women [47,48]. Taking into account this background, we propose H6:

**Hypothesis 6 (H6):** *Gender influences entrepreneurial intention.*

Entrepreneurial Parents

Family background plays an important role in the factors that encourage entrepreneurial intention, since they provide a favorable environment for the individual, as they are children, for developing new entrepreneurial ideas and motivating them to carry out the idea [49]. There is also empirical evidence that entrepreneurs are primarily descended from families in which one of the parents works via self-employment or on their own [50]. Therefore, regarding the entrepreneurial family environment, we formulate the following hypothesis:

**Hypothesis 7 (H7):** *Entrepreneurial parents influence the entrepreneurial intention of their university student children.*

Work Experience

The individual's employment situation or previous professional experiences influence the development of their entrepreneurial intention, showing that people who have worked as an employee have a greater interest in creating their own business [48,51]. One of the reasons for this is that it is easier for them to develop valuable contacts in the business environment [20], in addition to the mere fact that working in a sector allows the detection of opportunities in it [50]. On the other hand, some unemployed people choose to create their own companies to access the labor market because they do not have better job options [52]. Therefore, we will formulate the following hypothesis:

**Hypothesis 8 (H8):** *University students with work experiences tend to develop greater entrepreneurial intention.*

### 2.2.4. Reasons for Entrepreneurship

The reasons for a certain attitude play an important role in the intentions and behaviors of individuals, and there is a strong relationship between entrepreneurial behavior and entrepreneurial motives [53].

The primary characterization of the motives of entrepreneurship would be necessity or opportunity. Entrepreneurship by necessity does not usually contribute significantly to countries' economies since, in general, they lack innovation and do not usually generate employment because they usually opt for the cost leadership strategy [5]. In addition, entrepreneurs by necessity are mainly concerned with setting up businesses that guarantee certain profitability, which is usually low, and have no risk [54].

In the case of the Chinese community, which constitutes business out of necessity, we find preferences for the opening of a bazaar, a restaurant, or a food establishment; that is, these are small businesses with low chances of failure, aimed at covering basic consumer needs (which implies a low-profit margin) and which do not need much staff and have a quick return on the initial investment.

Among the reasons for the opportunity, we can find independence, desire for wealth, and need for achievement [55]. There are also other reasons for the undertaking, such as scientific knowledge, availability of resources, incubator organization, and social environment [56].

Among all these reasons for entrepreneurship, Antolín has pointed out one especially relevant for the Chinese community coming to Spain to start a business: the opportunity [57]. According to the GEM report, since 2015, the percentage of individuals who start a business by opportunity has been growing, concerning those who start a business by necessity [5].

We formulated the following hypothesis and relationship about the reasons for entrepreneurship:

**Hypothesis 9 (H9):** *The university student with entrepreneurial motives tends to develop a greater entrepreneurial intention.*

*Opportunity (independence, desire for wealth, and need for achievement) is the main reason for entrepreneurship by university students.*

Considering the hypotheses formulated, we offer our model for assessing the entrepreneurial intentions of university students, which can be seen in Figure 2.

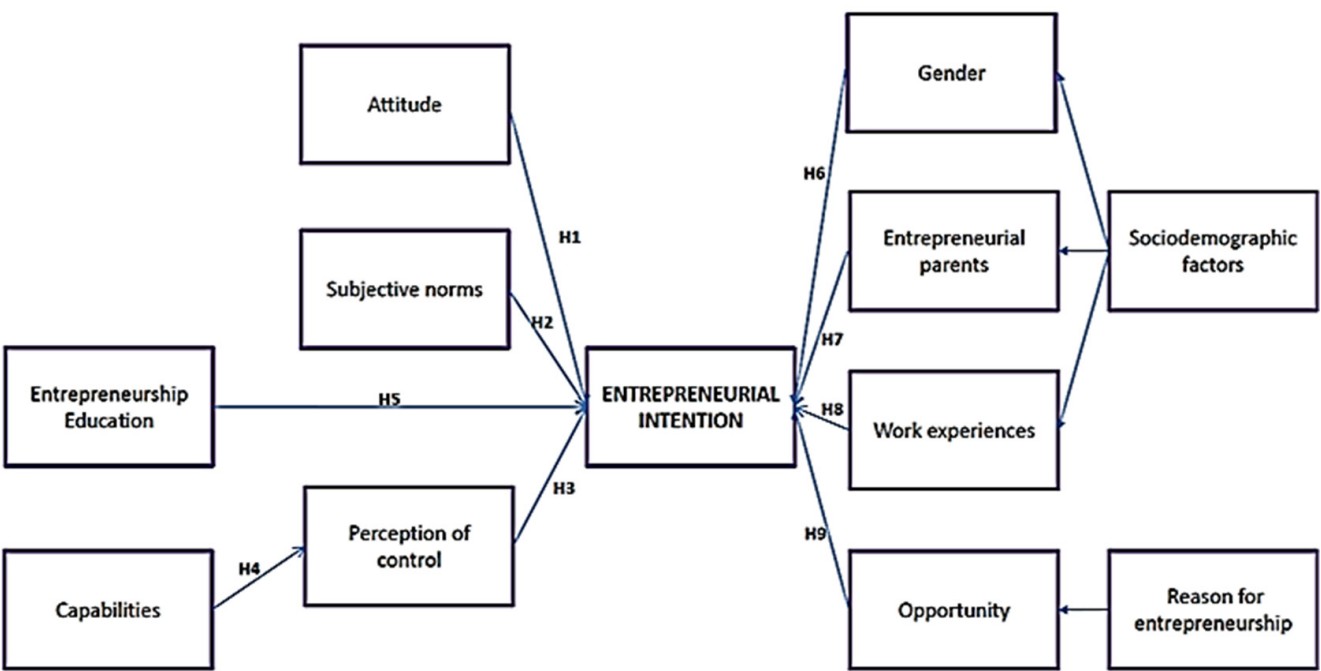

**Figure 2.** Model for studying entrepreneurial intentions.

### 3. Materials and Methods

*3.1. Methodology*

The methodology for the study of the sample is divided into two parts. The first was a descriptive analysis of the data. It was chosen to first perform a descriptive analysis of the data as, since these provide a basis for conducting future statistical analysis, it is also the best method for collecting data that describe the relationships and present the real world, offering a broad overview of the subject of study, in addition to helping the understanding of a given topic and the interpretation of results of more complex statistical models [58].

Secondly, to validate the hypotheses formulated, an analysis of the data obtained in surveys disseminated online to Chinese students present in universities in the community of Madrid was carried out using structural equation modeling (SEM).

SEM is a statistical measure that combines multiple regression and factor analysis increasingly used in scientific studies in the field of social sciences of academic research [59]. Survey data in social sciences often have many variables, and this model is commonly used to analyze these data [60]. As an advantage, this model would allow us to individually assess the interrelationship between multiple variables and control the level of reliability of each relationship. It would also allow us to study the relationships between latent variables (unobservable constructs) using observable variables. All of this would allow us to identify the relevant hypotheses, discarding those that were not supported by any empirical evidence [61]. The statistical analysis was performed using the partial least squares (PLS) method with the SmartPLS 3 software [62]. This methodology was chosen because it is exploratory analysis, since this sample had never been used before and, in addition, it was based on a proprietary model for the study of entrepreneurial intention, in which many different relationships had been considered together. The main disadvantage of the SEM analysis is that it requires a large sample size (around 200 individuals) or 10 cases per latent variable studied. As our work consisted of 136 respondents and 9 latent variables, to some extent, this limitation could be overcome [63].

### 3.2. Sample

The sample consisted of 136 Chinese students studying at universities in the community of Madrid. As a consequence of COVID-19, both students who were receiving classes in person and blended or remote learning were considered. According to data published by the Ministry of Inclusion, Social Security and Migration, at the end of 2020, there were a total of 1691 Chinese students with authorization to stay for studies in force in universities in the community of Madrid, which represents a significant drop compared with previous years, caused by COVID-19 [64]. The methodology used was closed surveys disseminated online through Google Forms from 9 January to 4 March, and the attitude questions were assessed using the Likert scale. The data collection was randomized, so it was a simple random sample. SmartPLS 3 software was applied to estimate the model. Of the 136 respondents, 5.9% had no entrepreneurial intention, 62.5% had entrepreneurial intention, 24.3% had already initiated actions for entrepreneurship, and 7.4% had already incorporated their own companies. The survey consisted of 20 questions to identify the profile of students, analyze the variables that influence entrepreneurial intention, and study the sociodemographic factors that influence entrepreneurial intention, as well as their motives for entrepreneurship.

In Table 1, we can see the different variables that affect entrepreneurial intention with their respective inspirations in the academic literature and the respective questions posed in the questionnaire for the study of the variable.

**Table 1.** Main literary reviews of the different variables that affect the entrepreneurial intention and questions raised in the questionnaire.

| Variable | Questions in the Survey | Literature Review |
|---|---|---|
| Attitude toward entrepreneurship | Is starting your own business an attractive idea for you? | [27,30,31,33,34] |
| Subjective norms | If you decided to start a company, would people around you approve of that decision? Indicate from 1 (strongly disagree) to 7 (strongly agree) as the case may be:<br><br>a.    Close family<br>b.    Close friends<br>c.    College classmates<br>d.    University professors<br>e.    Other people important to you | [27,29,36] |

**Table 1.** *Cont.*

| Variable | Questions in the Survey | Literature Review |
|---|---|---|
| Perception of behavior control | Please assess the degree of assent to the following statements:<br><br>a.　Starting a business and keeping it running would be easy for me.<br>b.　I am ready to start a viable company.<br>c.　I can control the process of creating a new company.<br>d.　I know the practical details needed to start a company.<br>e.　I know how to develop an entrepreneurial project.<br>f.　If you tried to start a business, you would have a high probability of subsistence. | [20,29,31] |
| Capabilities | Do you think you have a satisfactory level of the following skills to be an entrepreneur? Rate from 1 (strongly disagree) to 7 (strongly agree):<br><br>1.　Recognition of opportunities<br>2.　Leadership and communication skills<br>3.　Development of new products and services<br>4.　Implementation of ideas<br>5.　Networking and professional contacts | [37,38] |
| Entrepreneurship education | Have you taken any courses on business creation? | [40,43–46] |
| Gender | Please indicate your gender. | [32,48] |
| Entrepreneurial parents | Is your father or mother an entrepreneur? | [50,51] |
| Work experience | Do you work or have you worked in any company? | [25,51–53] |
| Reasons for entrepreneurship | Please indicate the degree of importance to you of the following reasons for entrepreneurship from 1 (strongly disagree) to 7 (strongly agree):<br><br>a.　Opportunity<br>b.　Need<br>c.　Independence<br>d.　The desire for wealth<br>e.　Need for achievement<br>f.　Need or interest in putting into practice the knowledge acquired in universities<br>g.　Availability of resources | [5,54,56,58] |

Surveys are the most widely used market research tools for obtaining primary information [65]. The survey was conducted online, as this allowed greater dissemination and reach in less time and at a lower cost, as well as allowing immediate processing of data [66]. The surveys were collected from 9 January to 4 March 2021, a period in which the first symptoms of COVID-19 began to appear in Madrid. One more reason we chose to use the online surveys disseminated through Google Forms was the geographical limitations caused by the pandemic. Attitude assessment questions were measured through the Likert scale, and these have been validated in previous studies [47,48,67].

## 4. Results

### 4.1. Descriptive Results

We reached 136 Chinese students studying at universities in the community of Madrid, of which the percentage of men and women surveyed were 52.2% and 47.8%, respectively. According to the studies being carried out by Chinese students, most of them were studying for undergraduate and master's degrees, with a percentage of 39% in both cases, 9% were in doctoral programs, and the remaining 13% were in courses outside those mentioned, such as language courses or their own degrees.

As for the stage of entrepreneurship the respondents were in, 63% had entrepreneurial intention, 24.3% had initiated actions for entrepreneurship, and 7% had their own companies. In comparison, the percentage of respondents without entrepreneurial intention was 6%.

Meanwhile, 87% of the respondents stated that they considered it an attractive idea to set up their own companies, which shows their positive attitude toward entrepreneurship.

Additionally, 83% of the respondents valued subjective norms at a value greater than 4; that is, they perceived that the people close to them would support them if they wanted to create a company. Within these, 34% valued it at 7, feeling sure that the people close to them would fully support them at the time of incorporation of a company.

The results also showed that 82% of the respondents perceived that they could control the process of creation and management of a company, valuing it at more than a 4, while 19% were sure that they could control the process of constitution and management of a company, valuing it at 7. The perception of control was determined by the capabilities of the respondents. Regarding these, 83% of the respondents had considered above a 4 their abilities to be an entrepreneur, while 34% valued it at 7, feeling fully capable of undertaking.

Half of the students surveyed considered that they had not received any course on entrepreneurship, while the other half claimed to have attended some course related to entrepreneurship. The most valued activities for the promotion of entrepreneurship were participation in existing projects, programs, or awards for young entrepreneurs (3.53 out of 7), contact or internships with local experts or entrepreneurs (3.23 out of 7), guided tours of companies, organizations, or other associations (3.23 out of 7), and preparation of a business plan (3.09 out of 7). At the same time, the least valued one was the realization of readings and works (2.45 out of 7).

The percentage of men and women with entrepreneurial intentions did not differ greatly, with the percentages being 53% and 47%, respectively.

The majority of the respondents, specifically 85%, had an entrepreneurial background. Among these, 44.2% of the respondents had entrepreneurial parents, 20.2% had entrepreneurial mothers, and 20.9% had both parents being entrepreneurial.

In addition, 86.8% of the respondents had work experience. Within these, 82.30% had worked in microenterprises (companies with less than 10 employees) and small companies (companies with less than 50 employees).

The most important reason for entrepreneurship for respondents was the desire for wealth (6 out of 7), followed by independence (5.97 out of 7), the need for achievement (5.95 out of 7), the perceived opportunity in the market (5.88 out of 7), the need or interest to put into practice the knowledge acquired in universities (5.87 out of 7), the availability of resources (5.85 out of 7) and finally, the need (5.76 out of 7).

### 4.2. Analysis of Models and Scenarios

The proposed model (Figure 3) was estimated by bootstrapping (5000 samples) using the Smart PLS 3 program and evaluated in two stages [68,69]. In the first stage, the measurement models (latent variables) were evaluated, and in the second stage, their structures were evaluated (relationship between the latent variables).

In the first stage in the measurement model, we found that loads of the standardized indicators with values greater than 0.70 and with a $p$-value $\leq 0.005$ translated into a strong significance of the indicators [70,71].

The adjusted $R^2$, which measured the degree to which the latent variables explained the behavior (variability) of the variable "Entrepreneurial Intention", took the value of 0.566, being acceptable in these types of social science research. For its interpretation, it should be noted that the closer it was to one, the better the fit of the model [72].

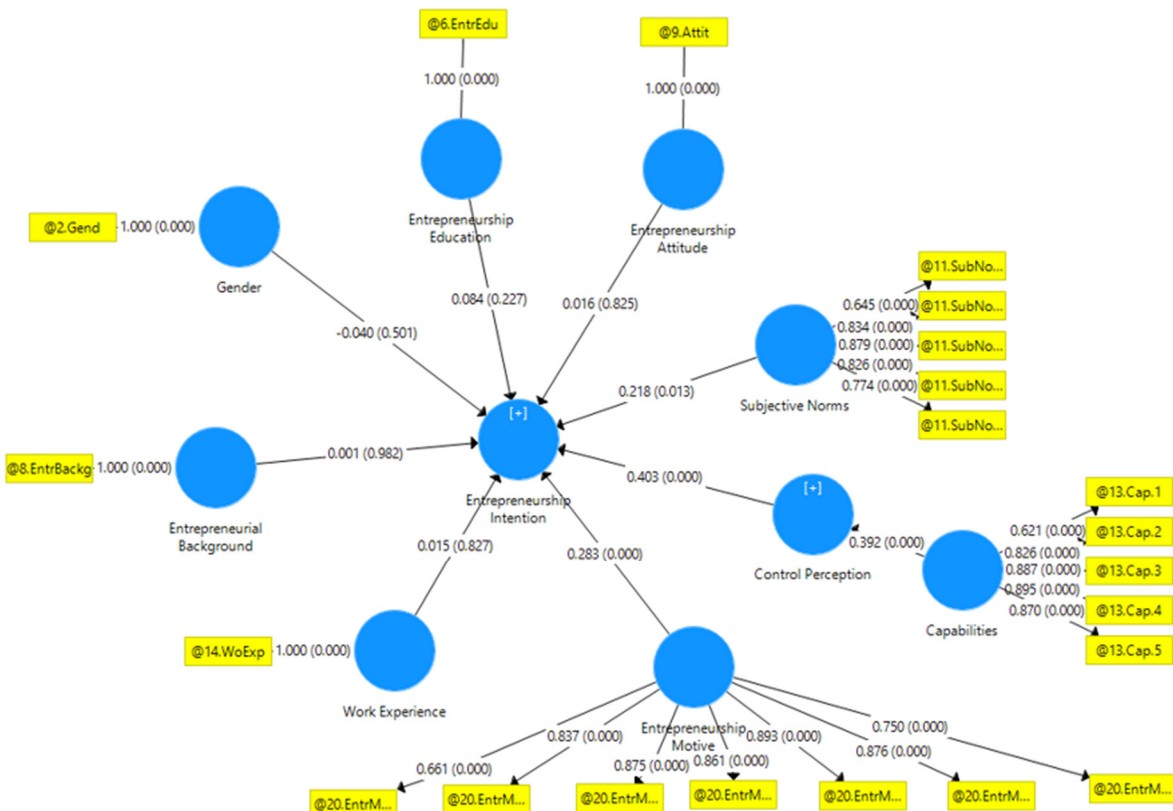

**Figure 3.** Statistical model of structural relationships of the different variables that affect the entrepreneurial intention.

The influence (on a normalized scale from 0 to 1) between the variable and the latent construct (entrepreneurial intention) and the acceptance or non-acceptance of the hypotheses are presented in Table 2.

**Table 2.** Acceptance of hypotheses according to standardized coefficients and *p*-value.

| Hypothesis | Standardized Coefficients | *p*-Value | Acceptance of Hypotheses |
|---|---|---|---|
| H1: Attitude → IE | 0.016 | 0.825 | Not accepted |
| H2: Subjective norms → IE | 0.218 | 0.013 | Accepted |
| H3: Perception of control → IE | 0.403 | 0.000 | Accepted |
| H4: Capabilities → Perception of control | 0.392 | 0.000 | Accepted |
| H5: Entrepreneurship education → IE | 0.084 | 0.227 | Not accepted |
| H6: Gender → IE | −0.040 | 0.501 | Not accepted |
| H7: IE' → entrepreneurial parents | 0.001 | 0.982 | Not accepted |
| H8: Work experience → IE | 0.015 | 0.827 | Not accepted |
| H9: Reason for entrepreneurship→ IE | 0.283 | 0.000 | Accepted |

Source: own elaboration.

The proposed model (Figure 1) performed the hypothesis contrast and reflected the measurement quality of the indicators on the latent variables, highlighting the significance of the form of measurement with a *p*-value < 0.000, as shown in Table 3.

**Table 3.** Significance of the relationship between the indicators and latent variable by standardized coefficients and *p*-value.

| Relation | | Standardized Coefficients | *p*-Value | Significance of the Relationship |
|---|---|---|---|---|
| **Latent Variable** | **Indicators** | | | |
| Reason for entrepreneurship | Need | 0.837 | 0.000 | Significant |
| | Independence | 0.875 | 0.000 | Significant |
| | Desire for wealth | 0.861 | 0.000 | Significant |
| | Need for achievement | 0.893 | 0.000 | Significant |
| | Need or interest in putting into practice the knowledge acquired in universities | 0.876 | 0.000 | Significant |
| | Availability of resources | 0.750 | 0.000 | Significant |
| Subjective norms | Close family | 0.645 | 0.000 | Significant |
| | Close friends | 0.834 | 0.000 | Significant |
| | College classmates | 0.879 | 0.000 | Significant |
| | University professors | 0.826 | 0.000 | Significant |
| | Other people important to you | 0.774 | 0.000 | Significant |
| Entrepreneurial capacity | Recognition of opportunities | 0.621 | 0.000 | Significant |
| | Leadership and communication skills | 0.826 | 0.000 | Significant |
| | Development of new products and services | 0.887 | 0.000 | Significant |
| | Implementation of ideas | 0.895 | 0.000 | Significant |
| | Networking and professional contacts | 0.870 | 0.000 | Significant |

Source: own elaboration.

H1: The attitude of the university student toward entrepreneurship is positively related to his or her entrepreneurial intention ($\beta = 0.016$, $p = 0.825$). This hypothesis was not accepted.

The relationship between the variables was positive and not significant. Higher scores in entrepreneurial attitudes were not related to higher entrepreneurial intent.

H2: The subjective norms of the university student toward entrepreneurship are positively related to his or her entrepreneurial intention ($\beta = 0.218$, $p = 0.013$). This was an accepted hypothesis.

The relationship between the variables was positive and significant. Higher scores on subjective norms were related to greater entrepreneurial intent.

H3: The perception of control of the behavior of the university student toward entrepreneurship is positively related to his or her entrepreneurial intention ($\beta = 0.403$, $p < 0.000$). This was an accepted hypothesis.

The relationship between the variables was positive and significant. Higher scores on the perception of control were related to greater entrepreneurial intent.

H4: The abilities of the university student are positively related to the perception of control of the university student ($\beta = 0.392$, $p < 0.000$). This was an accepted hypothesis.

The relationship between the variables was positive and significant. Higher scores on capabilities allowed for a greater perception of control.

H5: Entrepreneurship education is positively related to entrepreneurial intention in university students ($\beta = 0.084$, $p = 0.227$). This hypothesis was not accepted.

The relationship between the variables was positive and not significant. Higher scores in entrepreneurship education were not related to higher entrepreneurial intent.

H6: Gender influences entrepreneurial intention ($\beta = -0.040$, $p = 0.501$). This hypothesis was not accepted.

The relationship between the variables was negative and not significant. Gender differences in adolescents did not influence the development of entrepreneurial intention.

H7: Entrepreneurial parents influence the entrepreneurial intention of their university student children ($\beta = 0.001$, $p = 0.982$). This hypothesis was not accepted.

The relationship between the variables was positive and not significant. Having an entrepreneurial background did not influence the development of a greater entrepreneurial intention.

H8: University students with work experiences tend to develop greater entrepreneurial intention. ($\beta = 0.015$, $p = 0.827$). This hypothesis was not accepted.

The relationship between the variables was positive and not significant. Having worked or not as an employee did not affect the development of a greater entrepreneurial intention.

H9: The university student with entrepreneurial motives tends to develop a greater entrepreneurial intention ($\beta = 0.283$, $p < 0.000$). This was an accepted hypothesis.

The relationship between the variables was positive and significant. Having entrepreneurial motives favored the development of entrepreneurial intention.

The main reason for entrepreneurship by university students was opportunity, namely through independence ($\beta = 0.835$, $p < 0.000$), a desire for wealth ($\beta = 0.873$, $p < 0.000$), and a need for achievement ($\beta = 0.862$, $p < 0.000$).

The first hypothesis that the positive attitude toward entrepreneurship influenced the development of a greater entrepreneurial intention would remain unanswered since the relationship between both variables was not significant, with a $p$-value of 0.825 (greater than 0.05).

The hypothesis that subjective norms, perception of control, and capabilities have a positive relationship with entrepreneurial intention was accepted with a high level of confidence ($p$-value $< 0.013$), and the influence ranged from 0.218 to 0.403.

The hypotheses that entrepreneurship education, gender, the existence of entrepreneurial parents, and the possession of work experience influence the development of entrepreneurial intention were not accepted, as they showed $p$-values higher than 0.01 (from 0.226 to 0.982) and standardized coefficients less than 0.7 (from $-0.026$ to 0.084), which indicated the minor influence of the dimensions on the latent variable (entrepreneurial intention).

The hypothesis that individuals with entrepreneurial motives tend to develop greater entrepreneurial intent was also accepted, with a $p$-value less than 0.000 and a positive standardized indicator of 0.284. In addition, opportunity (independence ($\beta = 0.835$; $p = 0.000$), desire for wealth ($\beta = 0.873$, $p = 0.000$), and need for achievement ($\beta = 0.862$; $p = 0.000$)) were the most important reasons for entrepreneurship, which was confirmed with a high level of confidence.

## 5. Discussion

In this paper, the relationship of different variables for entrepreneurial intention were studied. The model studied and presented was partially aligned with the literature review, with Hypotheses 2, 3, 4, and 9 being accepted and Hypotheses 1, 5, 6, 7, and 8 not being accepted. The results show that the accepted hypotheses were validated with a high level of significance ($p < 0.05$).

H1: Although the background empirically validated that the positive attitude of the university student toward entrepreneurship has a positive relationship with the entrepreneurial intention [17,27], in our paper, the study showed the opposite. This may be because perhaps we should have taken into account other factors such as desirability, since according to the entrepreneurship potential model, attitude together with subjective norms forms desirability, and the latter positively affects the entrepreneurial intention of individuals, or perhaps another type of different metric should have been proposed for the assessment of the entrepreneurial attitude [35].

H2: It has been empirically demonstrated that subjective norms—that is, the perception of support of people important to the individual—have a positive relationship with

entrepreneurial intention [28,34]. This is in line with the results of our model, but our study differs in terms of the study sample, as we targeted Chinese students studying at universities in the community of Madrid. In addition, the indicators proposed for the measurement of the latent variable (subjective norms) were broader, since the previous works indicated taking only the indicators of close family, close friends, and other important people for the individual, while in our model, when dealing with university students, two additional indicators were raised regarding the perception of support of university professors and university colleagues. Therefore, we observed the literature on subjective norms having a positive impact on the development of entrepreneurial intention being due to the results obtained in our model for Hypothesis 2.

H3: The hypothesis that the perception of behavior control is positively related to the entrepreneurial intention of the university student is aligned with the study of Fayolle and Gailly in their work "The impact of entrepreneurship education on entrepreneurial attitudes and intention: hysteresis and persistence", in which they analyzed entrepreneurial intention and its antecedents to know if the entrepreneurship education program on entrepreneurial intention was effective, reaching the result that the perception of control was enhanced through the transfer of knowledge in universities and therefore caused an increase in entrepreneurial intention [38]. The indicators proposed for the measurement of the latent variable coincided with those offered by Linan and Chen, in which they applied Ajzen's theory of planned behavior for the study of the entrepreneurial intention of individuals from two different countries (Spain and Taiwan), reaching the result that attitudes and the perception of control are the variables that have the greatest influence on intention on entrepreneurs in Taiwan [1,27]. Therefore, Hypothesis 3, stating that the perception of control favors the creation of entrepreneurial intention, coincided with the background collected in the literature review.

H4: The perception of control is fundamentally determined by the capabilities that individuals believe they possess. In this paper, the capacities had a positive and very significant relationship with the perception of control, so H4 was consistent with the previous literature [36,37].

H5: As for entrepreneurship education, there is no doubt that it is positively related to entrepreneurial intention [41]. Entrepreneurship education is key to the success of entrepreneurial activity, since entrepreneurs often have to face problems that need multidisciplinary training [39].

However, according to the results obtained in this paper, entrepreneurship education is not related to entrepreneurial intention. That could be because in Spain, the training in entrepreneurship that entrepreneurs receive is not oriented toward training them in entrepreneurship; that is, it is not well designed for training in entrepreneurship, a problem that can be seen in the GEM report, in which experts valued at a 5.1 out of 10 the effectiveness of entrepreneurial education and training in the post-school stage, occupying fifth place in terms of obstacles to entrepreneurship [5]. The results of our model reflect this problem of the ineffectiveness of entrepreneurship programs (H5), so this factor needs to be improved to strengthen entrepreneurship in Spain.

In the same line, according to a study carried out by Delai where the determining factors of entrepreneurial intention in university students were analyzed, it was concluded that entrepreneurship education was not related to entrepreneurial intention, which could indicate the low effectiveness of the entrepreneurship programs that are being developed in universities for the promotion of entrepreneurial intention and the development of the entrepreneurial spirit and, with it, the need for curricular restructuring, giving greater importance to practice and going beyond the elaboration of a business plan, bringing students as close to the environment as possible to offer them the necessary experience to carry out the entrepreneurial process and offer them advice, support, and help for entrepreneurship [73]. In addition, there is also the need to design activities focused on strengthening the capacity to create and achieve goals, or encouraging them to think

differently and innovatively, which translates into risk and putting business ideas into practice [73–75].

H6: Regarding sociodemographic factors, even though according to all the GEM reports there is still a tendency for men to present a greater entrepreneurial intention compared with women, our paper shows that entrepreneurial intention does not differ in terms of the gender of the university student, which is in line with the study carried out by Morales, who found that, in the adolescence stage, gender does not influence the intentions of students, nor does it generate a different perspective of the business world, with the results on the entrepreneurial intentions of men and women being very similar [76]. Supporting this idea, we found several papers in which it was also shown that entrepreneurial intention did not differ in terms of the gender of the individuals [77,78].

H7: The existence of entrepreneurial parents should facilitate the development of entrepreneurial intention by the individual, since they provide them with a favorable environment from childhood to the development of entrepreneurial ideas and receive greater support to carry it out [49]. However, according to a study carried out by Díaz, Hernández, and Barata, in which they studied the entrepreneurial intentions of Spanish and Portuguese students, the hypothesis that the existence of a family entrepreneurial background did not influence the entrepreneurial intention was validated [79]. Our paper is in line with this author, since the existence of entrepreneur parents did not seem to relate with the entrepreneurial intentions of university students. Along the same line, the study carried out by Urbano, in which support organizations and attitudes toward entrepreneurial activity were studied, concluded that the entrepreneurial intention in individuals with entrepreneurial parents was lower than in those who worked for others [34]. We also found the study carried out by Franco, Haase, and Lautenschläger in which the influence of close entrepreneurial relatives on the development of students' entrepreneurial intentions was not appreciated [80]. In the same area, several studies started from the review of the literature that individuals with entrepreneurial parents tended to develop a greater entrepreneurial intention and obtained adverse results from their previous ideas [41,48].

H8: Work experience should have a positive impact on the development of entrepreneurial intention, since it facilitates the development of valuable contacts in the business environment and the fact that working in a sector allows the detection of opportunities for new business ideas [50,51]. However, our results showed the opposite. This could be because, in the case of university students, it was considered that most of them did not have enough work experience to positively influence their entrepreneurial intentions [47]. In the case of universities in the community of Madrid, in the undergraduate curriculum and in that of some master's degrees, only in the last semester was there a program of "external practices" to provide work experiences to the university, putting them in contact with companies. In addition, given the status of the student, most of them could only work in temporary employment options, part-time, and a small number of companies, so it was considered that the sample of our study did not reach the work experience necessary to influence the development of entrepreneurial intention [47]. However, Aponte and Gómez concluded that work experience does not influence the development of entrepreneurial intention [2].

H9: As for the motives of entrepreneurship, they play an important role in the intentions and behaviors of individuals, and a strong relationship between entrepreneurial behavior and entrepreneurial motives is recognized [53]. In this sense, our work is aligned with the review of the literature, since there was a very significant relationship between the motives of entrepreneurship and entrepreneurial intention (H9). Among the reasons for entrepreneurship, opportunity is the most important for the constitution of a company, and this includes independence, a desire for wealth, and a need for achievement [55,57]. Consistent with what was published by the GEM report, in which since 2015 there has been a considerable increase in the percentage of individuals who start a business by opportunity compared with those who start a business by necessity, our model is also in the same line, highlighting the importance of entrepreneurship by opportunity [5].

## 6. Conclusions

The results of this research show the relevance of the different variables studied toward entrepreneurial intention, an important indicator of the future behavior of entrepreneurship.

The positive attitude toward entrepreneurship, unlike what was reflected in the previous literature, did not have a significant relationship with the entrepreneurial intention, leaving H1 unvalidated.

According to the literature review, in this paper, we validated that those subjective norms and the perception of control of entrepreneurial behavior do have a significant relationship with the entrepreneurial intention of university students (H2 and H3).

In addition, we can confirm, according to the results, that the perception of behavior control is fundamentally determined by the abilities and capabilities possessed by the university students (H4).

As for the programs to promote entrepreneurship taught at universities in the community of Madrid, they did not seem to have a significant impact on the development of entrepreneurial intention (H5). The activity with the lowest value was the realization of readings and works, and the most valued one was the participation in existing projects, programs, or awards for young entrepreneurs and the contact or practice with experts or entrepreneurs in the same location.

Gender in adolescence did not influence the development of greater or lesser entrepreneurial intention (H6).

The existence of enterprising parents (H7) and previous work experience (H8) also did not seem to have significance with entrepreneurial intention.

As for the motives of entrepreneurship, it had a significant relationship with entrepreneurial intention (H9).

As we can see, half of the hypotheses were not accepted. This could be because we applied the entrepreneurial intention study model to a completely new sample, especially with different cultural aspects. The Chinese community is said to have an entrepreneurial culture, an aspect that can be reflected in the fact that 93.4% of our respondents showed entrepreneurial intention, had initiated actions for entrepreneurship, or even had already established their own companies.

More than half of the Chinese community is self-employed [81], while in Madrid, the figure of self-employed people is 14% [82]. This is due to the fact that the Chinese community has specific characteristics in which their entrepreneurial potential is manifested, such as entrepreneurship with equity. China is approaching parity in terms of the gender of entrepreneurs, as it continuously launches financial and business aid to women with entrepreneurial initiatives. As for their own funds, very few of the companies created in China require public funding, which greatly reduces the fear of failure due to the loss of control of the company. For innovative thinking, most of the companies created by the Chinese community are companies that offer technology-based goods and services and are very concerned with recognizing new consumer needs and doing their best to satisfy them. Regarding professional training, most Chinese entrepreneurs have a postgraduate education, which equips them with the knowledge and skills necessary for entrepreneurship. As for hard work, long working hours is a characteristic feature of Chinese entrepreneurs.

The systematic search for entrepreneurial opportunities in the market through the identification of consumer needs and the availability of resources are the aspects that characterize the entrepreneurial culture. To develop these capabilities, students need to be better educated to learn to cope with changes in the environment with flexibility through experience and knowledge and to reduce their fear of failure by establishing entrepreneurial support agencies, such as business incubators.

Some characteristics defend entrepreneurs, such as efficiency, responsibility, commitment, creativity, innovation, and risk-taking, so skills such as creativity, strength, will, autonomy, confidence, and resilience are the ones that should be reinforced through the curriculum for the development of new entrepreneurs.

As the results show, entrepreneurial intention is not conditioned by the gender of the Chinese community, so efforts should also continue to be made to decrease the gender gap of entrepreneurs through information and government programs of support, guidance, and funding led by women, as they will help to enhance economic growth and improve the lives in society. Similarly, the government should promote entrepreneurship, especially in industries with high growth potential and little competition, such as technology companies. The state should also create a more suitable environment for innovative entrepreneurship by reducing barriers, strengthening public services, and encouraging university students to set up new companies. The collaboration of financial and banking entities is also considered necessary to facilitate obtaining financing, such as channels to support SMEs.

As for the limitations of the study, it should be pointed out first that this is a preliminary and exploratory analysis, so it is difficult to extrapolate the results at this time and draw comprehensive conclusions. Moreover. entrepreneurial intention is a difficult subject to analyze, since it depends on personal issues and people are very complex, different from one another, and difficult to predict, so all the works carried out by different authors to study the entrepreneurial intentions of individuals differ to a greater or lesser extent.

In addition, there has not been an in-depth analysis of the variables that influence entrepreneurial intentions, such as questions regarding psychological factors that affect the entrepreneurial intentions of individuals and the obstacles to entrepreneurship.

Because of geographic limitations caused by COVID-19, the desired number of respondents could not be reached. In addition, due to the pandemic, the number of Chinese students in Spanish universities were significantly reduced in the 2020–2021 academic year, making it very difficult to access them online, so the results obtained were scarce and unrepresentative of the sample to be analyzed. The sample should be increased to achieve more generalized results and more comprehensive and representative conclusions.

It would also be interesting to make a comparison between Spanish university students and Chinese students in terms of their entrepreneurial profiles and intentions.

Moreover, this study confirmed that Chinese students tend to have a very high entrepreneurial intention, and half of them are inclined to set up their own companies in Spain and establish commercial relations with the country, so it would be interesting to study the reasons why the Chinese community chooses Spain for entrepreneurship.

As for practical implications, we provided a study model for entrepreneurial intention applicable to different interest groups, such as at the academic level, which will allow us to identify students with entrepreneurial potential to inform them about the aid and support programs for entrepreneurs and the resources and services available to reduce the level of failure of the entrepreneurial process, as well as motivate them to formalize and put into practice the business ideas they have in mind. It also allows us to understand the role of the variables studied in the dynamization of entrepreneurship and the reasons for the increase in companies that generate social and environmental economic value.

The identification of the most important training activities for the promotion of entrepreneurship allows the design of a more effective study plan for the development of entrepreneurial intention in university students.

Through research, it has been shown that the vast majority of Chinese students tend to have a strong entrepreneurial intention concerning the Spanish population, and knowing the aspects that determine the entrepreneurial culture of China allows the state and the government to design new policies for the creation of a more suitable environment for entrepreneurship.

**Author Contributions:** Conceptualization, S.L., C.D.-P.-H., J.L.M.B. and C.L.-L.; data curation, S.L. and J.L.M.B.; formal analysis, S.L. and J.L.M.B.; investigation, S.L. and C.D.-P.-H.; methodology, S.L., C.D.-P.-H. and J.L.M.B.; software, J.L.M.B.; supervision, C.D.-P.-H. and J.L.M.B.; validation, S.L., C.D.-P.-H., J.L.M.B. and C.L.-L.; visualization, C.D.-P.-H., J.L.M.B. and C.L.-L.; writing—original draft, S.L.; writing—review and editing, C.D.-P.-H. and J.L.M.B. All authors have read and agreed to the published version of the manuscript.

**Funding:** This article has been financed with funds from OPENINNOVA High Performance Research Group (URJC-V921).

**Institutional Review Board Statement:** Not applicable.

**Informed Consent Statement:** Informed consent was obtained from all subjects involved in the study.

**Data Availability Statement:** The data presented in this study are available on request from the corresponding author.

**Acknowledgments:** We wanted to give our sincere thanks to our friends and colleagues at the university, who have helped us conduct and disseminate the surveys. We also want to thank Rey Juan Carlos University, who offered the ideas and the resources to start with our work.

**Conflicts of Interest:** The authors declare no conflict of interest; The funders had no role in the design of the study; in the collection, analyses, or interpretation of data; in the writing of the manuscript, or in the decision to publish the results.

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
