# Peer review of "Entrepreneurial Intention of Chinese Students Studying at Universities in the Community of Madrid"

_sustainability, doi:10.3390/su14095475_

Round 1

Reviewer 1 Report

The article submitted to «Sustainability» journal, but it not reflected sustainability.
Review of literature on entrepreneurial intention has not been conducted well. This is because the authors haven't included in more recent papers on this specific field of research. The autors should do more review of more current literature and add it to the discussion in this paper.  I would see the following changes:
- A more detailed analysis of the existing research gaps.
- A more detailed analysis of the existing literature on the subject area (scientific literature from the last 5 years). The sources from Clarivate Analytics Web of Science and/or Scopus are very welcome. Aditionally, the autors should elaborate more about what academic contributions have been made by this paper. 
The methodology: This section should also identify strengths and weaknesses of the methodology.
The conclusion should discuss the obtained results with direct and special stress on the value-added and policy implication. Include more limitations and future research directions.
Moreover, the editorial and  language quality of the paper should be improved. 

Author Response

Dear Reviewer,

Thank you very much for your comments and suggestions!

Please find attached a document in response to your comments.

We hope we have interpreted your comments correctly and have made the appropriate changes.

Please let us know if we need to make any other modifications.

Thank you

Best regards

Reviewer 2 Report

The paper concentrates on the important issue from the perspective of the entrepreneurial intention of students;

Discussion and Conclusions should be better. The results are presented clearly, but the findings are not compared and contrasted with relevant literature. Linking theoretical considerations with empirical findings and providing some planning insights is critical in a journal with the scope of Sustainability;

Another issue of the paper is its novelty. What are the new findings of the study as there is much work done on the same issue?

Author Response

(The authors gave the same response as above.)

Reviewer 3 Report

Thank you for the opportunity to contribute to improvement of this manuscript. The paper is well-written and easy to follow. There are a few parts of the manuscript which require further development and improvements. The paper appears to be a replication of dozens of researches testing the very well-known TPB model, with no clear original contribution.

  • By the end of introduction, the fundamental research questions (in addition to objectives) that the authors try to understand deserve development and argumentation in close connection with the context presented.

  • The sampling methodology is not explained. Did the authors use probabilistic or non-probabilistic sampling?
  • Is the number of valid answers (136) enough for the research? What is the total population of Chinese students studying in the Community of Madrid? What are the Margin of error and the Confidence level?

  • When was the data collected and how? This should be explained in the Methodology.
  • The authors are running a regression model (not explained in the Methodology) and an exploratory factor analysis. This must be defined and explained in the Methodology section. 

  • Fig. 2: Instead of presenting the well-known TPB, the authors should introduce the research conceptual model, which displays the relationships between variables tested.
  • How do you explain that only half of the research hypotheses were validated? 
  • The theoretical value of the research should be clearly outlined. What are the novel and unique contributions to the related literature? The paper should be more suggestive in terms of scientific relevance and contribution that the author(s) want to bring to the field.
  • What are the practical implications of the research results?
  • As there are some clear research limitations (only one being mentioned in the paper), the authors should define future avenue of research.

Good luck with further improvement of the manuscript!

Author Response

(The authors gave the same response as above.)

Round 2

Reviewer 1 Report

Thanks for the improvements made.
Doubt - article was submitted to Sustainability but does not reflect sustainability.

Reviewer 3 Report

Thank you for the improvements done. Good luck with publication of your paper!